# Tetracycline Residues in Bovine Muscle and Liver Samples from Sicily (Southern Italy) by LC-MS/MS Method: A Six-Year Study

**DOI:** 10.3390/molecules24040695

**Published:** 2019-02-15

**Authors:** Gaetano Cammilleri, Andrea Pulvirenti, Antonio Vella, Andrea Macaluso, Gianluigi Maria Lo Dico, Vita Giaccone, Vincenzo Giordano, Massimo Vinciguerra, Nicola Cicero, Antonello Cicero, Giuseppe Giangrosso, Stefano Vullo, Vincenzo Ferrantelli

**Affiliations:** 1Istituto Zooprofilattico Sperimentale della Sicilia “A. Mirri”, via Gino Marinuzzi 3, 90129 Palermo, Italy; antonio.vella@izssicilia.it (A.V.); andrea.macaluso@izssicilia.it (A.M.); gigilodico@gmail.com (G.M.L.D.); vita.giaccone@gmail.com (V.G.); vincenzo.giordano@gmail.com (V.G.); antonellocicero@gmail.com (A.C.); g.giangrosso1@gmail.com (G.G.); stefano.vullo@izssicilia.it (S.V.); vincenzo.ferrantelli@izssicilia.it (V.F.); 2Dipartimento Scienze della Vita, Università degli studi di Modena e Reggio Emilia, Via Università 4, 41121 Modena, Italy; andrea.pulvirenti@unimore.it; 3Dipartimento di Prevenzione Veterinario, ASP 1 Agrigento, Viale della Vittoria 321, 92100 Agrigento, Italy; massimovinciguerra@gmail.com; 4Dipartimento SASTAS, Università degli studi di Messina, Polo Universitario dell’Annunziata, 98168 Messina, Italy; ncicero@unime.it

**Keywords:** LC-MS/MS, veterinary drugs, tetracycline, risk assessment, validation process, bovine samples

## Abstract

We examined a total of 369 bovine liver and muscle samples for the detection of oxytetracycline (OTC), tetracycline (TC), chlortetracycline (CTC), and doxycycline (DOX) residues by implementation and validation of a LC-MS/MS method. The method showed good recovery values between 86% and 92% at three levels of concentrations. The linearity tests revealed r^2^ > 0.996 for all the tetracyclines examined. Furthermore, the Youden test revealed that the method was robust. Only 14.4% of the samples showed OTC and TC residues in a concentration range of 10.4–40.2 µg kg^−1^. No CTC and DOX residues were found in all the samples analyzed. Liver samples showed the highest average values (31.5 ± 20.6 and 21.8 ± 18.9 for OTC and TC, respectively). The results showed a low incidence of TCs in all the samples examined, in comparison with other studies reported in the literature. A significant decrease in TC residues frequency was found from 2013 (*p* < 0.05). This work reports for the first time epidemiological data on the presence of TC residues in liver and muscle samples of cattle farmed in Sicily (Southern Italy). The very low incidence of TC residues indicates a continuous improvement in farming techniques in Southern Italy, which is essential to ensure consumers’ protection.

## 1. Introduction

Antibiotics are widely used substances in animal farming. Among the antibiotics, tetracycline is the most used antimicrobial substances for the prevention of bacterial disease of farmed animals, followed by macrolide, lincosamides, penicillins, and sulphonamides [1,2]. Tetracyclines are one of the cheapest classes of antibiotics available, and their cost in real terms is declining due to improved manufacturing technology [3]. Some tetracyclines such as chlortetracycline, oxytetracycline, tetracycline, and demethylchloretetracycline have animal origins, whereas doxycycline, methacline, minocycline, and meclocycline were obtained semi-synthetically.

The mechanism of action of tetracyclines is bacteriostatic and consists of the inhibition of bacterial protein synthesis. They are also able to chelate some divalent cations (Ca^2+^), causing indirectly the inhibition of numerous bacterial enzymatic systems [4]. 

Tetracyclines are high-spectrum antibiotics against Gram-positive and Gram-negative bacteria, aerobic and anaerobic microorganisms, mycoplasmas, spirochetes, actinomycetes, chlamydias, and some protozoa. Based on the broad spectrum of action, tetracyclines are used in all localized and systemic bacterial infections and in prophylaxis and the treatment of viral infections. In particular, they are used in cases of septicemia, infections of the gastrointestinal tract and respiratory tract, and in the prophylaxis and therapy of bovine methritis [4].

These antibiotics are added directly to feed or water or can be administered in aerosols. The improper use of tetracycline, especially in subtherapeutic doses, leads specific strains of *Staphylococcus, Coliforms, Bacillus*, and *Clostridium* to develop a resistance [3,4].

The human exposure to subchronic levels of tetracycline may lead to gastrointestinal disturbances and hypersensitivity [5,6]. One of the most frequent complications of tetracycline oral treatment is the appearance of dismicrobism. In humans, serious forms of hepatotoxicity have been recorded at therapeutic doses of tetracyclines [7]. Furthermore, the intake of tetracyclines by pregnant women has led to a higher incidence of abortions, premature births, and malformations of the fetus [4,8].

There are definitive tetracycline, oxytetracycline, chlortetracycline maximum residue levels (MRL) for all productive livestock species and for all foodstuffs (muscle, kidney, liver, milk, and eggs). Doxycycline cannot be used in animals that produce milk and eggs for human consumption.

Liquid chromatography is the officially recognized analytical method for the determination of tetracycline residues in the muscle of cattle, pigs, sheep, goats, and horses [9,10,11].

At present, very little is known about the accumulation of these substances in bovine muscle and liver samples farmed in Southern Italy. In this work, we examined for the first time in the literature the presence of four tetracyclines residues (oxytetracycline, tetracycline, chlortetracycline, doxycycline) in muscle and liver samples of cattle raised in Sicily (Southern Italy) by an LC-MS/MS method in order to obtain a six year study on the occurrence of these substances in Southern Italy farms.

## 2. Results

The validation data are listed in Table 1. The linearity calculation showed values of linear correlation coefficient (r^2^) higher than 0.996 for all the analytes examined. Excellent results were obtained for accuracy and precision parameters of intra-day and inter-day analyses with relative standard deviation (RSD) values within 10%. The trueness values obtained were in the range of 86–92%. The limits of detection (LOD)s were found to be 2.0–4.0 µg kg^−1^ and the limits of quantification (LOQs) 6.0–12.0 µg kg^−1^ for TCs in muscle and liver. The decision limit (CCα) and decision capability (CCß) were in the range of 119.2–128.5 and 118.0–126.8 µg kg^−1^ for muscle and 303.1–305.9 and 305.6–309.4 µg kg^−1^ for liver, respectively. The application of the Youden test revealed that the tested factors were not critical.

Out of 369 muscle and liver samples analyzed during this work, 53 (14.4%) had detectable levels for tetracycline residues (Figure 1). Of these, tetracycline residues were mostly found in liver samples (9.7%). No chlortetracycline and doxycycline residues were found in all the samples analyzed. The tetracycycline groups that comprised the positive samples were oxytetracycline, which was found in 34 samples (9.2%), and tetracycline (5.1%). The positive samples with mean and range values are shown in Table 2. The liver samples revealed the highest concentrations (39.7 µg kg^−1^ and 36.7 µg kg^−1^ for oxytetracycline and tetracycline, respectively) with range values between 20 and 39.7 µg kg^−1^. The bovine samples from Siracusa showed the highest incidence of tetracycline residues (six out of the 21 samples analyzed in this province, equal to 28.6%), followed by Ragusa (19.4%) and Catania (16.3%). No tetracycline residues were found in muscle and liver samples from Enna province.

The annual analysis of the presence of tetracycline residues verified a unimodal distribution, with a peak in tetracycline use during 2013 and a constant decrease until 2015 for all the sample types analyzed. However, only liver samples showed significant differences in tetracycline residues between years of sampling (Kruskal–Wallis chi-squared = 18.415, *p* = 0.002468).

## 3. Materials and Methods

### 3.1. Reagents and Materials

Methanol, acetonitrile, and formic acid 99.9% (LC-MS grade) were supplied from VWR (VWR International PBI Srl Milan, Italy). Purified water was obtained in the laboratory using a Milli-Q system. Disodium ethylene-diaminetetraacetate (Na_2_EDTA) was obtained from Sigma Aldrich (Sigma-Aldrich, Milan, Italy). Standards of oxytetracycline (OTC), tetracycline (TC), chlortetracycline (CTC), doxycycline (DOX), and demeclocycline (DEMC) were purchased from Sigma-Aldrich (Sigma-Aldrich, Milan, Italy). Demeclocycline was used as an internal standard (IS) because it is an obsolete antibiotic.

### 3.2. Standard Solutions

OTC, TC, CTC, DOX, and DEMC stock standard solutions (1.0 mg/mL) were prepared every six months by dissolving in methanol and stored at −20 °C. The mixed standard solution (100 ng/mL) was diluted to volume with mobile phase and stored in an amber tube at −20 °C.

### 3.3. Sample Collection and Extraction

A total of 369 bovine muscle and liver samples were collected in 2010 to 2015 from monitoring and surveillance programs on a random basis (Table 3). All the samples were stored at +4 °C and transported to the Residues Laboratory of the Istituto Zooprofilattico Sperimentale della Sicilia. Each sample was homogenized and stored at −20 °C until the time of analysis.

About 5.0 g of the homogenized samples were accurately weighed and mixed with 10 mL of methanol in a 50 mL Falcon tube. Subsequently, 200 µl of EDTA 0.1M was added, and the solutions were vortexed for 1 min and centrifuged at 3500 rpm for 10 min. Then, 4 mL of the extract was filtered and evaporated under nitrogen stream at +40 °C. The residue was then dissolved in 1ml of mobile phase.

### 3.4. Instrumentation

The analysis was performed on a Thermo Fischer UHPLC system (Thermo Fisher Scientific, California, U.S.A.) consisting of an ACCELA 1250 quaternary pump and an ACCELA autosampler.

A Thermo Scientific Hypersil Gold reversed-phase UHPLC column (50 mm, 2.1 mm ID, 1.9 μm) was used for the chromatographic separation. The LC eluents were water (A) and acetonitrile (B), containing 0.1% (v/v) formic acid. The gradient started with 95% eluent A for 1.0 min, continued with linear variation to 10% A in 6.0 min; these conditions were maintained for 3.0 min. The system returned to 95% A in 0.5 min and was re-equilibrated for 5 min. The column temperature was 30 °C, and the sample temperature was kept at 6 °C. The flow rate was 0.4 mL min^−1^ and the injection volume was 5 μL.

The mass spectrometer was a triple quadrupole TSQ Vantage (Thermo Fisher Scientific, California, CA, USA) in positive electrospray ionization mode (ESI). The product ion scans of each analyte were performed by direct infusion (10 µL min^−1^) of 1 mg L^−1^ individual standard solutions with the built-in syringe pump through a T-junction, mixing with the blank column eluate (200 µL min^−1^).

The ESI parameters optimized were as follows: capillary voltage 4.5 kV; capillary temperature 310 °C; vaporizer temperature 150 °C; and sheath and auxiliary gas pressure were fixed at 40 and 15 (arbitrary unit), respectively. The collision gas was argon at 1.5 mTorr, and peak resolution of 0.7 FWHM was used on Q1 and Q3. The scan time for each monitored transition was 0.02 s, and the scan width was 0.02 *m*/*z*. The collision energy parameters associated with the precursor and the product ions are given in Table 4. Acquisition data were recorded and elaborated using Xcalibur^TM^ version 2.1.0.1139 software from Thermo.

### 3.5. Validation Procedure

The method was validated according to the procedure for residues in food animal products described by the EU Commission Decision 2002/657/EC under Council Directive 96/23/EC (Official journal, 2002). Specificity, linearity, recovery, limit of detection (LOD), limit of quantification (LOQ), decision limits (CCα) and the detection capability (CCß), precision (repeatability and the within-laboratory reproducibility), and accuracy of the method were determined for the validation. 

Blank samples were fortified at three different concentrations in equidistant steps (50.0–100.0–150.0 μg kg^−1^ and 150–300–450 μg kg^−1^ for muscle and liver samples, respectively). Ten spiked samples, at each of the three levels, were analyzed. The thirty-replicate analysis was repeated on three separate days, giving 90 independent determinations. The specificity of the method was assessed using blank samples and spiked samples. Representative chromatograms were generated to show that the extraneous peaks are resolved from the peaks for tetracycline. 

For the linearity, standard curves were determined at five levels in the range of 6.0–350 µg/L, including zero. A regression model was applied to the calibration data set, and correlation coefficients r^2^ > 0.995 were considered acceptable.

Precision is expressed as the percent relative standard deviation (RSD%) of concentrations calculated for spiked samples. Trueness was expressed as recovery rates by calculating three concentration levels (LOQ, MRLs and 2 MRL) in triplicate, according to Blasco et al. (2009) [9]. The recoveries were assessed by comparing the peak area of the concentration measured to the peak area of the spiked concentration.

The limit of detection (LOD) was calculated at a signal-to-noise ratio of 3, whereas the LOQ value was calculated by using a signal-to-noise ratio of 10.

CCα was established by the following steps: 20 blank samples of bovine liver and muscle were analyzed, and the signal-noise ratio (S/N) was calculated at the time window in which the analyte was expected. CCα values were defined as three times of S/N. CCß was calculated by analyzing 20 blank samples spiked at CCα, and then the CCα value plus 1.64 times the corresponding standard deviation was equal to CCß.

The method was tested for ruggedness by means of the Youden robustness test [12]. The study was conducted at 100 μg kg^−1^ for muscle and 300 μg kg^−1^ for liver samples, and six operating factors were chosen: time of centrifugation, time of stirring, autosampler temperature, methanol concentration of the extraction solution, centrifugation rate, and stirring rate. 

### 3.6. Data Collection and Statistical Analysis

All the tetracycline levels were expressed as µg kg^−1^. The results under the LOQ of the method were considered for the statistical analysis as half of the LOQ values, according to Helsel (2005) [13]. 

The conditions of normal distribution and homogeneity of variances of the data had not been met; therefore, a Kruskal–Wallis test was carried out to evaluate tetracycline content differences between years of sampling. The statistical analysis was carried out with the R^®^ 3.0.3 software.

## 4. Discussion

The LC-MS/MS method proposed and validated in this work has proven to be effective and reliable for all the tetracyclines examined giving satisfactory validation data. The validation results obtained in this work are comparable with other methods reported in the literature [9,14]. Furthermore, the method proposed has proven to be very fast (50 min/per sample, including the chromatographic run) and robust, according to the Youden approach. 

To the best of our knowledge, this was the first time that a method for the detection of tetracycline residues in bovine muscle and liver samples was tested for ruggedness. The experimental design resulted in the conduction of experiments relative to the selection of six variables that were chosen during the sample preparation and analysis. The application of this test consisted in the introduction of minor simultaneous changes in these parameters according to an established experimental design, with the aim of identifying the critical factors that have to be controlled in order to obtain accurate assay results. This study confirmed that the tested factors are not critical, confirming that the method proposed gave reliable results during the six years of monitoring.

A very low incidence of the presence of tetracycline residues was found during the six years of monitoring, with very low concentrations of oxytetracycline and tetracycline.

The results obtained showed a reduced use of tetracyclines in beef production of Sicily, probably due to the low number of animals bred in the territory compared to the large world production [6,15] and the constant increase in the agricultural area utilized due to the consequent decrease of the animals to be reared [16]. This condition has led many producers to adopt extensive rearing systems with natural forage, reducing the need for the use of veterinary drugs. The censuses on the regional bovine production report in fact a prevailing decrease in the number of animals reared between 2010 and 2017 (366,015 cattle for 2010 and 48,489 for 2017), with a slight spike of regrowth between 2012 and 2013 that corresponds to the higher incidence of tetracycline residues found in this work (42% of the total positive samples).

The highest prevalence of tetracycline residues in the samples from Siracusa and Ragusa seems to be related to the highest number of cattle raised and the largest average size of cattle farms in comparison with the other provinces [16].

Unfortunately, our method was carried out and validated before the adoption of the Commission Regulation 37/2010 (22 December 2009); therefore, we decided to carry out a method for the sole detection of the parent compounds, in accordance with other studies reported in the literature.

The tetracycline levels found in this work are comparable with those found by Vragovic et al. (2011) [17] in 75 meat samples from Croatia by ELISA method and up to 52 times lower than those found in bovine meat samples from Germany [15] and Kenya [6] by HPLC method, confirming the reduced use of these substances in this area. It is, however, relevant to consider the detection methods for reported frequency of detection of tetracycline residues in bovine muscle samples because of the differences in sensitivity and specificity of such tests.

The results of this study also confirmed the greater capacity of tetracycline to accumulate in the liver compared to muscle [4,18]. This is due to the fact that tetracyclines are distributed uniformly into the tissues and are found in high concentrations in the excretory organs, especially the liver [19,20]. Given their high affinity with plasma proteins (50–70% for oxytetracycline and chlortetracycline, respectively), tetracyclines undergo extensive enterohepatic circulation, which leads to prolongation of their elimination half-lives. Residual depletion studies have shown that following oral administration of therapeutic doses of oxytetracycline, appreciable concentrations of the drug are not observed in bovine, edible tissues five days after treatment [4].

The widespread use of tetracycline has become a serious problem since they are present as residues in animal products intended for human consumption that can promote the occurrence of antimicrobial-resistant bacteria and be toxic for humans. The tetracycline concentrations found in this six-year study were up to 50 times lower than what was reported in the literature, indicating a continuous improvement in the rearing techniques on bovine farms of Southern Italy, which is essential to ensure consumers’ protection. Nevertheless, continuing monitoring is certainly required to ensure efficient consumer protection.

## 5. Conclusions

The present work wanted to give an exhaustive overview about the presence of TCs in bovine muscle and liver samples farmed in Southern Italy (Sicily). A very fast and reliable LC-MS/MS method was validated for the purpose. The results showed a very low incidence of TCs in the samples examined. Analogous studies on the TC assessment of random samples report generally higher incidences with higher concentrations. Milk is an important source of protein, vitamins, and minerals [21,22], but its nutritional contribution to a balanced diet can be compromised by the high presence of toxic drugs such as tetracyclines. To the best of our knowledge, the present work gives for the first time in literature epidemiological data on the presence of TC residues in liver and muscle samples of cattle farmed in Sicily. Our findings indicate a continuous improvement in farming techniques in Southern Italy in order to improve food safety.

## Figures and Tables

**Figure 1 molecules-24-00695-f001:**
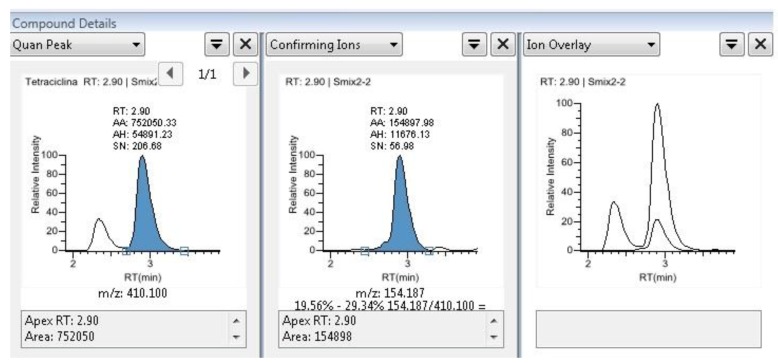
Chromatograms of a sample examined for TCs with relative m/z ratios.

**Table 1 molecules-24-00695-t001:** Accuracy and precision of the method for tetracyclines in spiked bovine samples analyzed at three concentration levels.

Sample Type	Compound	Spiked Level (µg kg^−1^)	Overall Recovery (%)	Intra-day RSD (%)	Inter-day RSD (%)
Liver	Oxytetracycline (OTC)	150	86.6	8.5	9.4
		300	90.4	7.6	8.9
		450	88.7	5.8	9.6
	Tetracycline (TC)	150	86.9	8.4	8.4
		300	91.5	6.3	7.8
		450	92	6.1	6.9
	Chlortetracycline (CTC)	150	88.4	5.9	6.7
		300	90.6	6.6	8.4
		450	91	6.7	8.2
	Doxycycline (DOX)	150	87.6	8	5.8
		300	86.3	9.4	8.2
		450	88.54	6.3	8.9
Muscle	OTC	50	91.6	5.3	7.5
		100	88.7	7.6	9.6
		150	90.5	8.3	9.9
	TC	50	86.4	9.4	6.4
		100	87.6	8.3	9.5
		150	87.2	7.7	8.7
	CTC	50	86	9.2	8.2
		100	88.4	7.2	9.4
		150	91.8	5.9	9.8
	DOX	50	90.9	8.5	9.5
		100	91	6.7	8.4
		150	85.5	5.9	7.9

**Table 2 molecules-24-00695-t002:** Mean, range, and frequency of positive samples for tetracyclines (expressed in µg kg^−1^) of bovine samples examined. LOQ: limits of quantification.

	N		>LOQ		Liver (Mean±SD)	Muscle (Mean±SD)	Liver (min–max)	Muscle (min–max)
	Liver	Muscle	Liver (%)	Muscle (%)				
OTC	152	217	25 (6.7)	9 (2.5)	31.5 ± 20.6	15.9 ± 10.4	23.9–40.2	15.0–28.6
TC	152	217	13 (3.6)	6 (1.6)	21.8 ± 18.9	10.9 ± 12.6	20.8–38.56	10.4–27.9
CTC	152	217	n.d.	n.d.	-	-	-	-
DOX	152	217	n.d.	n.d.	-	-	-	-

**Table 3 molecules-24-00695-t003:** Number of bovine liver and muscle samples analyzed for the detection of tetracyclines sorted by sampling years.

	2010		2011		2012		2013		2014		2015	
	Liver	Muscle	Liver	Muscle	Liver	Muscle	Liver	Muscle	Liver	Muscle	Liver	Muscle
Catania	15	14	11	9	8	10	8	8	8	7	13	11
Agrigento	1	-	0	2	-	2	4	6	3	8	-	1
Messina	-	-	3	3	1	3	6	8	3	9	1	2
Palermo	-	1	-	-	12	11	8	9	5	8	5	5
Enna	-	-	-	-	-	1	1	7	-	1	1	0
Trapani	-	-	-	-	-	-	5	6	4	9	1	1
Caltanissetta	-	-	-	-	1	3	5	4	5	4	-	-
Siracusa	1	1	-	1	-	-	3	3	3	9	-	-
Ragusa	2	1	2	2	2	2	7	6	3	10	-	-

**Table 4 molecules-24-00695-t004:** Precursor and most abundant product ions and their optimal collision energy. IS = internal standard.

	Parent	Product 1 (CE)	Product 2 (CE)
Tetracycline (TC)	445.1	154.2 (27)	410.1 (18)
Oxytetracycline (OTC)	461.1	426.3 (20)	443.2 (13)
Chlortetracycline (CTC)	479	444.1 (22)	462.1 (18)
Doxycicline (DOX)	445.1	428.1 (17)	409.7 (49)
Demeclocycline (IS)	465.1	448.2 (20)	392.7 (28)

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
