# Peer review of "Tetracycline Residues in Bovine Muscle and Liver Samples from Sicily (Southern Italy) by LC-MS/MS Method: A Six-Year Study"

_molecules, 2019, doi:10.3390/molecules24040695_

Round 1
Reviewer 1 Report
The manuscript presents an interesting study on the occurrence of TCs in ~400 bovine samples collected in Sicily, during 6 years and at different locations. The analytical effort invested in this study is noteworthy, and results presented may be very useful for analysts and veterinary personal working on the topic.
However, I do have a few questions and comments for the authors:
- A chromatogram of a positive/spiked sample is required
- In Introduction, the term "risk assessment" must be deleted, as risk assessment studies shall include risk evaluation. This is a study of residues "occurrence".
- In section 2.3, sample collection and extraction, tissues were pooled??? Did authors mean samples were ground/homogenized? Pooling implies mixing different samples. Please clarify.
- In section 2.6, what about samples below LOD?
- Results, first paragraph, line 162, include units for liver (ug kg-1)
- have authors considered the possibility of merging Results and Discussion?
- With regard to MRLs for TCs, in the particular case of OTC, TC and CTC, EU Regulation 37/2010 established limits for the sum of parent drug and its 4- epimer. Authors must clarify this fact in the text, and discuss why 4-epimers have not been considered in their study.
Author Response
Dear reviewer
We have carefully revised our manuscript. We have attached a word file with the track changes made to ease your perusal of our manuscript changes.
All the step by step changes are reported below:
1. A chromatogram of a positive/spiked sample is required
We provided a chromatogram of a positive/spiked sample as a figure in the main document.
2. In Introduction, the term "risk assessment" must be deleted, as risk assessment studies shall include risk evaluation. This is a study of residues "occurrence"
We apologise for this mistake. The sentence was replaced by “in order to obtain a 6 years study on the occurrence of these substance in Southern Italy farms”.
3. In section 2.3, sample collection and extraction, tissues were pooled??? Did authors mean samples were ground/homogenized? Pooling implies mixing different samples. Please clarify.
We apologise for the mistake. We intended that each sample was only homogenized and stored at -20°C. We reformulated the sentence in the main document.
4. In section 2.6, what about samples below LOD?
According to Helsel (2005), we decided to consider for statistical analysis only the results below the LOQ of the method because the LOQ combines fitness of purpose criteria with statistical criteria, unlike the LOD.
5. Results, first paragraph, line 162, include units for liver (ug kg-1)
We added this information in the main text.
6. Have authors considered the possibility of merging Results and Discussion?
We apologise for that but we decided to divide results and discussion sections in accordance to the authors’ guideline.
7. With regard to MRLs for TCs, in the particular case of OTC, TC and CTC, EU Regulation 37/2010 established limits for the sum of parent drug and its 4- epimer. Authors must clarify this fact in the text, and discuss why 4-epimers have not been considered in their study.
We apologize for this inconvenience. Unfortunately, our method was carried out and validated 5 months before the adoption of the Commission Regulation 37/2010 (22 december 2009), therefore we decided to carry out a method for the sole detection of the parent compounds, in accordance with other studies reported in literature (Muriuki et al. 2001; Korner et al. 2001; Vragović et al. 2011). We validated a LC-MS/MS method for the detection of the TCs metabolites in 2012, so we have results of the metabolites detection of the samples examined in this work from 2012 to 2015. However, we decided to report only the parent compound results in order to have a reliable statistical comparison between years of sampling, given that the principal aim of this work is to give a multiannual study on the presence of these substances. We decided to remove any citations regarding the Commission Regulation 37/2010 in order to be consistent with the aim of the manuscript.
Hope these changes could be helpful for the manuscript reconsideration.
King regards
Reviewer 2 Report
Introduction - lines 39 - 41 and 48 - 50 - citation are missing;
line 155 - Results are point 3 no 2
line 121 - limit of detection and limit of quantification should be written in lowercase letters
line 169 - wrong name from tetracyclines "tetracycilines
I think the authors should add Conclusion chapter, because now missing.
Author Response
Dear Reviewer
We have carefully revised our manuscript. We have attached a word file with the track changes made to ease your perusal of our manuscript changes.
All the step by step changes are reported below:
1. Introduction - lines 39 - 41 and 48 - 50 - citation are missing;
We added these information as suggested.
2. line 155 - Results are point 3 no 2
We have made this change according to your suggestion
3. line 121 - limit of detection and limit of quantification should be written in lowercase letters
We have made this change according to your suggestion
4. line 169 - wrong name from tetracyclines "tetracycilines
We have made this change according to your suggestion
5. I think the authors should add Conclusion chapter, because now missing.
A conclusion chapter was added to the main text according to your precious suggestion.
Hope these changes could be helpful for the manuscript reconsideration.
King regards
Reviewer 3 Report
This is a well written work, Nevertheless it has some very important issues that must be cleared by the authors before recommending its publication.
First, at the beginning of the manuscript, authors state that they are going to obtain a 6 years risk assessment on the presence of this substance in Southern Italy farms. However, there is no formal “risk assessment” methodology presented, or results associated to a RA. So, is clear that this was not an objective of the study so it must be removed from the manuscript. Authors can state that they will perform a statistical analysis the data.
The most important issue, and my main concern about this work is that they analyze 5 tetraciclines. However, they did not analyze the right marker compound. Is well known that for TC, OTC and CTC the sum of the parent compound and the metabolite must be analyzed. Furthermore, this is stablished in European legislation COMMISSION REGULATION (EU) No 37/2010, that is cited by the authors.
In this work, this fact was not considered and the metabolites were not included either in the validated methodology or in the samples analysis.
Specific comments can be found in the text attached.

Author Response
Dear Reviewer
We have attached a word file with the track changes made to ease your perusal of our manuscript changes.
All the step by step changes are reported below:
1. First, at the beginning of the manuscript, authors state that they are going to obtain a 6 years risk assessment on the presence of this substance in Southern Italy farms. However, there is no formal “risk assessment” methodology presented, or results associated to a RA. So, is clear that this was not an objective of the study so it must be removed from the manuscript. Authors can state that they will perform a statistical analysis the data.
We apologise for this mistake. We have introduced this terminology wrongly. The sentence was replaced by “in order to obtain a 6 years study on the occurrence of these substance in Southern Italy farms”
2. The most important issue, and my main concern about this work is that they analyze 5 tetraciclines. However, they did not analyze the right marker compound. Is well known that for TC, OTC and CTC the sum of the parent compound and the metabolite must be analyzed. Furthermore, this is stablished in European legislation COMMISSION REGULATION (EU) No 37/2010, that is cited by the authors.
We apologize for this inconvenience. Unfortunately, our method was carried out and validated 5 months before the adoption of the Commission Regulation 37/2010 (22 december 2009), therefore we decided to carry out a method for the sole detection of the parent compounds, in accordance with other studies reported in literature (Muriuki et al. 2001; Korner et al. 2001; Vragović et al. 2011).
3. Lines 65-66 I do not quite agree with this afirmation. There is information recently publish about tetraciclines in tissues. Our group have published some work in that , also Berendensen et al and Anadon et al. I think thata the value of this work is the study thruogh six years. Erase this sentance.
We replace this sentence with “At present, very little is known about the accumulation of these substances in bovine muscle and liver samples farmed in Southern Italy”
4. Did you measure de sum of the parental compound and the active metabolite as established in the EU legislation? If that so, why you didnt include the standards.
Unfortunately, we validated a LC-MS/MS method for the detection of the TCs metabolites in 2012, so we have results of the metabolites detection of the samples examined in this work from 2012 to 2015. However, we decided to report only the parent compound results in order to have a reliable statistical comparison between years of sampling, given that the principal aim of this work is to give a multiannual study on the presence of these substances. We decided to remove any citations regarding the Commission Regulation 37/2010 in order to be consistent with the aim of the manuscript.
5. How long does the analisys takes? is well know that tetraclines stability is limited, even at -20°C.
The LC-MS/MS analysis of all the samples examined was carried out max 6 hours after the preparation, in order to have reliable data.